# External Validation of COVID-19 Risk Scores during Three Waves of Pandemic in a German Cohort—A Retrospective Study

**DOI:** 10.3390/jpm12111775

**Published:** 2022-10-28

**Authors:** Lukas Häger, Philipp Wendland, Stephanie Biergans, Simone Lederer, Marius de Arruda Botelho Herr, Christian Erhardt, Kristina Schmauder, Maik Kschischo, Nisar Peter Malek, Stefanie Bunk, Michael Bitzer, Beryl Primrose Gladstone, Siri Göpel

**Affiliations:** 1Department of Internal Medicine 1, University Hospital Tübingen, Otfried-Müllerstrasse 10, 72076 Tubingen, Germany; 2Department of Mathematics and Technology, University of Applied Sciences Koblenz, Joseph-Rovan-Allee 2, 53424 Remagen, Germany; 3Medical Data Integration Center (meDIC), University Hospital Tübingen, Schaffhausenstraße 77, 72072 Tubingen, Germany; 4Institute of Medical Microbiology and Hospital Hygiene, University Hospital Tübingen, Elfriede-Aulhorn-Str. 6, 72076 Tubingen, Germany; 5Clinical Research Unit for Health Care Associated Infections, German Center for Infection Research (DZIF), Otfried-Müllerstrasse 10, 72076 Tubingen, Germany

**Keywords:** COVID-19 risk score, NEWS, COVID-GRAM, 4C-Mortality score, pandemic, ICU

## Abstract

Several risk scores were developed during the COVID-19 pandemic to identify patients at risk for critical illness as a basic step to personalizing medicine even in pandemic circumstances. However, the generalizability of these scores with regard to different populations, clinical settings, healthcare systems, and new epidemiological circumstances is unknown. The aim of our study was to compare the predictive validity of qSOFA, CRB65, NEWS, COVID-GRAM, and 4C-Mortality score. In a monocentric retrospective cohort, consecutively hospitalized adults with COVID-19 from February 2020 to June 2021 were included; risk scores at admission were calculated. The area under the receiver operating characteristic curve and the area under the precision–recall curve were compared using DeLong’s method and a bootstrapping approach. A total of 347 patients were included; 23.6% were admitted to the ICU, and 9.2% died in a hospital. NEWS and 4C-Score performed best for the outcomes ICU admission and in-hospital mortality. The easy-to-use bedside score NEWS has proven to identify patients at risk for critical illness, whereas the more complex COVID-19-specific scores 4C and COVID-GRAM were not superior. Decreasing mortality and ICU-admission rates affected the discriminatory ability of all scores. A further evaluation of risk assessment is needed in view of new and rapidly changing epidemiological evolution.

## 1. Introduction

COVID-19 spread around the world in alarming speed, burdening healthcare systems and hospitals. A large number of COVID-19 patients in life-threatening conditions had to be treated in hospitals that were provisionally set up, often by less experienced staff, indicating the need for an easy and objective clinical scoring model to identify high-risk patients [1]. Furthermore, risk scores are the first step in personalizing medicine to stratify individual patients to medical treatment. Several COVID-19-specific risk scores have been developed and validated during the first wave involving different populations and clinical settings: The COVID-GRAM risk score (*GRAM*: Guangzhou Institute of Respiratory Health Calculator at Admission) was developed in a large cohort of 1590 patients from 575 hospitals in China [2] with the aim of predicting critical illness in COVID-19 patients defined as admission to ICU (intensive care unit), mechanical ventilation, or death. The risk score summarizes 10 variables based on a logistic model and requires—due to the complexity—an online tool for usage ([2], see Appendix B). In 2020, the 4C-Mortality Index (Coronavirus Clinical Characterisation Consortium) was developed by including 260 hospitals and 35,463 patients in England, Scotland, and Wales to predict in-hospital mortality [3]. In contrast to the younger patient cohort (mean age of 49 years) with a lower mortality rate of 3.2%, which helped develop the COVID-GRAM risk score, the 4C-Mortality Index was developed in an older cohort (mean age was 73 years) with a higher in-hospital mortality rate of 32.2% [3]. Such differences question the generalizability of these scores and warrant further validation of the scores in different populations and various clinical settings. Moreover, several preexisting risk scores developed for other infectious diseases and intensive care medicine such as NEWS (National Early Warning Score), CRB-65 (confusion, respiratory rate, blood pressure—age 65), and qSOFA (quick sequential organ failure assessment) are also in use, though their predictivity among COVID-19 patients remains unclear. For example, during the study period, NEWS was routinely being used as a risk assessment tool to initiate a timely clinical response such as nurse–doctor–contact or notification of ICU for COVID-19 patients at the University Hospital Tübingen. Though several authors examined the predictive performance of some of the scores in COVID-19 risk stratification [4,5,6,7,8,9,10], the performance of multiple scores has not been compared in the same patient population. Over the course of the pandemic, epidemiological circumstances and therapeutic options rapidly changed: remdesivir received conditional approval in the treatment of COVID-19 patients in July 2020, and the benefit of dexamethasone was proven by the RECOVERY Collaborative Group [11]. In December 2020, new SARS-CoV-2 variants with high transmissibility and a potential immune escape were reported in the United Kingdom and South Africa, which were declared by the WHO as variants of concern [12]. In 2021, the European Union licensed several COVID-19 vaccinations, which had shown high efficacy and safety in clinical trials. The progress of immunization and vaccination, different COVID-19 variants, various healthcare systems, and the increasing efficacy of treatment have had a great impact on COVID-19 mortality risk. Nevertheless, there is a residual risk reported for critical illness even among the vaccinated population [4]. Until today, there is no risk score recommended in the German COVID-19 guidelines [13].

Amidst this background, a retrospective cohort study was conducted with the primary objective of evaluating the performance of common clinical scores (NEWS, qSOFA, and CRB-65) and COVID-19-specific clinical scoring models (COVID-GRAM, 4C-Mortality score) for the prediction of ICU admission and in-hospital mortality and comparing their predictive performance in a cohort of hospitalized COVID-19 patients in Germany. To the best of our knowledge, this is the first study to compare all of the abovementioned scores in a defined population, comparing their discriminative abilities in different waves during the course of the pandemic.

## 2. Materials and Methods

### 2.1. Study Design and Setting

We conducted a monocentric retrospective cohort study at the University Hospital Tübingen (tertiary care hospital) located in Tübingen, Germany, during the first three waves of the COVID-19 pandemic.

### 2.2. Study Population

The study recruited patients ≥18 years admitted to the university hospital due to COVID-19 between 1 March 2020 and 30 May 2021. SARS-CoV-2 infection was confirmed by positive real-time polymerase chain reaction (RT-PCR). Exclusion criteria were as follows:Patients not hospitalized for COVID-19 disease;Patients with a patient decree determining a DNR/DNI (do not resuscitate/do not intubate) situation;Patients transferred to our ICU from other hospitals, for example, due to the need for extracorporeal membrane oxygenation (ECMO).

### 2.3. Definition of Cohorts

The study period was divided into three cohort periods according to the classification of the waves by the Robert-Koch Institute [14] and based on key epidemiological factors: the number of COVID-19 cases, prevalence of different SARS-CoV-2 variants in Germany and at the University Hospital Tübingen, and availability of standardized specific therapy and vaccination. The prevalence of virus variants was assessed according to own data when genotyping was available and general evidence [12,15]. COVID-19 vaccination rates in Germany were assessed according to the results of the COVIMO study group [16]. The study population was thus divided into cohort 1 from 1 March to 30 June 2020, cohort 2 from 1 July 2020 to 7 March 2021, and cohort 3 from 8 March to 30 May 2021 (see Table 1). We decided to slightly deviate from the Robert-Koch Institute’s classification of the COVID-19 waves due to the decreasing incidence of hospital admissions in summer (only 7 patients fulfilled the inclusion criteria between July and September 2020).

### 2.4. Data Collection and Score Validation

We retrospectively collected clinical, demographical, and outcome data for each cohort by using the clinical information and documentation systems. Comorbidities included were chronic respiratory disease, cardiovascular disease, chronic liver disease, chronic kidney disease, HIV (human immunodeficiency virus) infection or AIDS (acquired immunodeficiency syndrome), organ transplantation, diabetes mellitus, malignancy, and chronic neurological conditions. Demographical and epidemiological data collected were age, sex, body mass index, COVID-19 vaccination, and DNR/DNI status. The scores studied were the Quick Sequential Organ Failure Assessment score (qSOFA), National Early Warning Score (NEWS), CRB-65 score, COVID-GRAM risk score, and 4C-Mortality Score (see Table 2). Each of the scores was separately calculated for each patient using the admission data. In case of missing values at admission, we decided to collect the earliest parameter on the day of admission or day 1 after admission to improve the power of the study. This included the laboratory parameter serum urea concentration and radiological data. In case of missing direct bilirubin, we used total bilirubin. Glasgow Coma Scale was retrospectively calculated (for the workflow, see Appendix A).

### 2.5. Statistical Analysis

After the inclusion of patients by the above-defined inclusion and exclusion criteria, in a second step, patients with missing values were excluded. In the case of at least one missing value, we did not calculate the specific score (Appendix A, for characteristics of excluded patients, see Appendix A). The primary outcomes were endpoints of critical illness defined by ICU admission and in-hospital mortality. Discriminative indices of the selected scores including sensitivity, specificity, and positive and negative predictive values were calculated. Confidence intervals were assessed via a bootstrapping method. The discriminative abilities of the scores were assessed and compared by using the area under the receiver operating characteristic curve (AU-ROC) and the area under the precision–recall curve (AU-PRC). Equality of the AU-ROCs was tested using DeLong’s method and a bootstrapping method as well [17,18]. We decided to show the bootstrap findings in the results. To correct for multiple testing, we used Bonferroni–Holm adjusted *p*-values. An adjusted *p*-value of 0.05 or less was regarded as significant. Data were analyzed by R Version 4.1.2 using the packages readxl, pROC, precrec, boot, and ggplot2 [19,20,21,22]. We reported continuous variables as mean with the first and third quartiles and categorical variables as a number with the percentage of the cohort. Continuous variables were compared using the Mann–Whitney U-test, and categorical variables were compared by the use of Fisher’s exact test.

## 3. Results

### 3.1. Study Population

Overall, 756 patients were admitted to the hospital with SARS-CoV-2 infection from February 2020 to May 2021, of which 347 (45.9%) patients were included (see Table 3). Of these, 134 (38.6%), 187 (53.9%), and 26 (7.5%) formed the first, second, and third cohorts, respectively. A total of 153 patients (44.1%) were women, and the average age at the time of admission was 65.4 years (IQR 57.0 to 78.0). A total of 173 (49.9%) showed COVID-19-specific findings in X-ray, 184 (53%) had dyspnea, 179 (51.6%) had cough, and 189 (52.2%) had fever. The most common comorbidities were cardiovascular diseases, hypertension, and diabetes. A total of 146 patients (42.1%) were treated with steroids, and 131 (37.8%) required specific medication with remdesivir. One (0.3%) participant was vaccinated at least once against COVID-19; the type of vaccine is unknown. The overall in-hospital mortality rate was 9.2% (*n* = 32), and 23.6% (*n* = 82) required ICU treatment. The characteristics of the study cohort are shown in Table 3.

### 3.2. Comparison of Cohorts

The ICU-admission and in-hospital-mortality rates significantly decreased from the first to the second wave (*p*-values of 0.002 and 0.044, respectively). The mean oxygen saturation at admission was significantly higher in the second cohort (*p*-value of 0.034), whereas documented fever decreased (*p*-value of 0.036). We did not find significant differences between the first and the second wave considering age, sex, and comorbidities. Due to the low sample size of the third-wave cohort, statistical tests were less significant. Only the mean respiratory rate was significantly lower in the third cohort compared with the first cohort (*p*-value of 0.04). Nevertheless, we could observe a decreasing trend in the ICU-admission rate, in-hospital mortality rate, and average age also between the second and third cohorts.

### 3.3. Predictive Performance of the Scores

We included the whole study population in the comparison of the predictive performance of the scores. Due to the low sample size of the cohort, three statistical tests for the comparison between cohort 3 and the other two cohorts had a lower power (Figure 1). Therefore, the third cohort was excluded from the comparison between the cohorts (discriminatory indices, Appendix A). Overall, the NEWS model performed the best regarding ICU admission, which was confirmed by ROC-AUC (0.83; CI 0.76–0.88), whereas the 4C-Score showed a higher PR-AUC (0.64; CI 0.50–0.78) than NEWS. We found significant differences in the ROC-AUC compared with qSOFA (0.70; CI 0.64–0.77); however, differences to the other models were not statistically significant. The 4C-Score had the highest ROC-AUC (0.81; CI 0.69–0.90) with regard to in-hospital mortality. Again, differences to the other scores were not statistically significant. qSOFA, CRB-65 score, and COVID-GRAM performed lower (Figure 2).

### 3.4. Comparison between the Cohorts

The performance of NEWS and 4C-Score was better in cohort 1 than in cohort 2. The NEWS had a ROC-AUC of 0.88 (CI 0.80–0.94) in the first cohort and 0.71 (CI 0.60–0.81) in the second cohort with regard to ICU admission, whereas the ROC-AUC was 0.75 (CI 0.62–0.86) in the first cohort and 0.73 (CI 0.53–0.89) in the second cohort with regard to in-hospital mortality. Differences were statistically significant between the cohorts concerning ICU admission (*p*-value of 0.011). However, differences were not statistically significant regarding in-hospital mortality. The 4C-Score showed a ROC-AUC of 0.84 (CI 0.73–0.93) in the first cohort and 0.58 (CI 0.42–0.72) in the second cohort concerning ICU admission. The ROC-AUC was 0.87 (CI 0.78–0.94) in the first cohort and 0.59 (CI 0.33–0.84) in the second cohort concerning in-hospital mortality. ICU admission differences between the cohorts were statistically significant (*p*-value of 0.002), as well as in-hospital mortality differences (*p*-value of 0.045). We did not find any significant difference within the cohorts regarding qSOFA, CRB-65, and COVID-GRAM (details in Appendix A).

## 4. Discussion

We evaluated the performance of various COVID-19-specific as well as commonly used risk scores in a retrospective cohort of COVID-19 patients over the course of three waves of the pandemic. In our study, COVID-specific scores including COVID-GRAM and 4C-Score failed to show significant superiority compared with the NEWS model. This was shown especially for the prediction of ICU admission. Liang et al. reported an AUC of 0.88 (CI 0.85–0.91) in the derivation cohort of COVID-GRAM for the composite endpoint of ICU admission, need for invasive ventilation, and death [2]. This was not well-reflected in our findings with AUC 0.75 for ICU admission and 0.65 for in-hospital mortality. Knight et al. reported an AUC of 0.79 (CI 0.78–0.79) in the derivation cohort of the 4C-Score for the endpoint in-hospital mortality [3]. In our first cohort—which might reflect the original derivation cohort— we found an AUC 0.84 for ICU admission and 0.87 for in-hospital mortality, thus confirming their results. The usage of COVID-GRAM and 4C-Mortality score in clinical assessment is more complex compared with NEWS using more variables and requiring radiological assessment in addition to the laboratory parameters as well as information on relevant comorbidities; COVID-GRAM needs to be calculated by a calculator available online. NEWS includes more vital signs than both the COVID-specific scores; the assessment can be easily made on the bedside. In every-day clinical life, this is a clear advantage. The different baseline characteristics (especially the average age, ICU admission rates, and mortality rates) of the derivation cohorts and differences in epidemiological conditions (e.g., healthcare systems, need for triage) might be the reason for differences of performance in our cohort. It has been demonstrated before that the impact of changing vital sign categories on prognosis in terms of mortality is larger in older patients [23]. This can be especially discussed for the derivation cohort of COVID-GRAM with a mean age of 48.9 years in comparison with our cohort with a mean age of 65 years. 4C performed better in our cohort than COVID-GRAM but did not show significant superiority to NEWS. Interestingly, we could not reproduce the AUC findings of the COVID-GRAM. This implies the importance of evaluating risk scores in different settings. Our findings are in line with a previous study conducted in Italy, which also described the differences between COVID-GRAM (AUC 0.785, CI 0.723–0.838), 4C-Score (AUC 0.799, CI 0.738–0.851), and NEWS (AUC 0.764, CI 0.700–0.819) as not statistically significant with regard to all-causes in-hospital death [7]. Another investigation figured out NEWS2 (an advanced version of the NEWS examined in this study, including hypercapnic respiratory failure instead of oxygen saturation) as prognosticating critical illness in COVID-19 better than COVID-GRAM (AUC of 0.87, CI 0.80–0.93 for NEWS2 and of 0.77, CI 0.68–0.85 for COVID-GRAM) [5]. CRB-65 and qSOFA as commonly used risk scores for pneumonia and sepsis were both clearly outperformed by NEWS—thus highlighting the COVID-specific importance of monitoring of oxygen saturation in addition to the respiratory rate possibly influenced by the silent hypoxemia that is characteristic for severe COVID-19 [24].

The in-hospital-mortality and ICU-admission rates significantly decreased during the study period comparing the three defined cohorts. Especially the ICU-admission rate in our cohort is comparable with national data—according to the data derived from a German federal hospital payment institute, the national ICU-admission rate of hospitalized patients was 30% in the first wave and 14% in the second wave [25]—in our cohort, 37% and 16%, respectively. The in-hospital mortality in our cohort was at 17% in the first and 5% and 0% in the second and third waves, respectively. In a study aggregating health insurance data of 158,490 German patients, the in-hospital mortality of the three waves was at 22.2%, 21.7% and 14.8% [26]. Of note, significant differences between hospitals were seen in this study.

One reason for the difference that can be discussed is a significantly lower mean age in our cohort than in the above-mentioned cohort (67 and 65 years versus 72 and 74 years in the first and second wave, respectively). Our hospital has a specific expertise in ARDS treatment, which might have modified the treatment outcome. For the third wave, patient numbers in our cohort were too low to provide statistically significant numbers.

A decrease in the in-hospital-mortality and ICU-admission rates has also been described in other countries, mostly interpreted as the composite effect of better treatment and the protective effect of immunization and vaccination within the population [27]. By the definition of the cohorts that we choose, this can also be assumed, investigating three different “waves” of patients with different treatment options and availability of vaccination. The effect of the vaccination rollout during cohort 3 is presumably reflected in the reduced average age of mostly unvaccinated hospitalized patients, since vaccination rollout was prioritized in the beginning for patients of higher age and with severe comorbidities. As the in-hospital-mortality and ICU-admission rates decreased, the discriminatory ability of the scores also decreased. Due to a small sample size, we did not calculate the AUC and PR-AUC of the third cohort, but it can be assumed that with further decreasing in-hospital-mortality and low-ICU-admission rates, the trend continues.

A further evaluation of risk assessment according to rapidly changing epidemiological circumstances is needed, especially considering new variants, treatment, and prevention options. According to our findings, a risk stratification of patients at admission could be recommended in Germany using the simple bedside score NEWS.

## 5. Limitations

Due to the retrospective nature of the study, we had to exclude patients with missing variables, thus possibly creating a bias toward more severely ill patients in which more parameters are documented. Furthermore, the sample size in the third cohort was respectably smaller than those in the other cohorts. The study was conducted at a university hospital (tertiary care hospital and center for extracorporeal membrane oxygenation), creating a further possible bias. Since the analysis was carried out until June 2021, the succeeding variants of concern Delta and Omicron are not reflected in our study. On the other hand, the hospitalization, mortality, and ICU-admission rates were steadily declining especially with the Omicron variants, so it can be assumed that the discriminatory ability of the scores decreases further. The strength of our study is a broad comparison of several widely used risk scores in a German population. Furthermore, we provide a comparison of the score performance spanning over the course of the pandemic with changing epidemiological and therapeutic conditions.

## 6. Conclusions

Interestingly, in our hospitalized cohort, the COVID-specific risk-scores COVID-GRAM and 4C were not superior to NEWS especially in predicting ICU admission, even though they use additional risk factors acknowledged for severe COVID-19 disease. As a simple bedside-use scoring system, NEWS has proven to be a useful tool to identify hospitalized patients at risk for critical illness in our study population. This finding is important especially in clinical real-life circumstances, and the usage of the NEWS score for the risk stratification of patients at hospital admission should be discussed. Decreasing mortality rates and ICU-admission rates affected the discriminatory ability of all scores, so further investigation will be needed to address the highly dynamic epidemiological evolution.

## Figures and Tables

**Figure 1 jpm-12-01775-f001:**
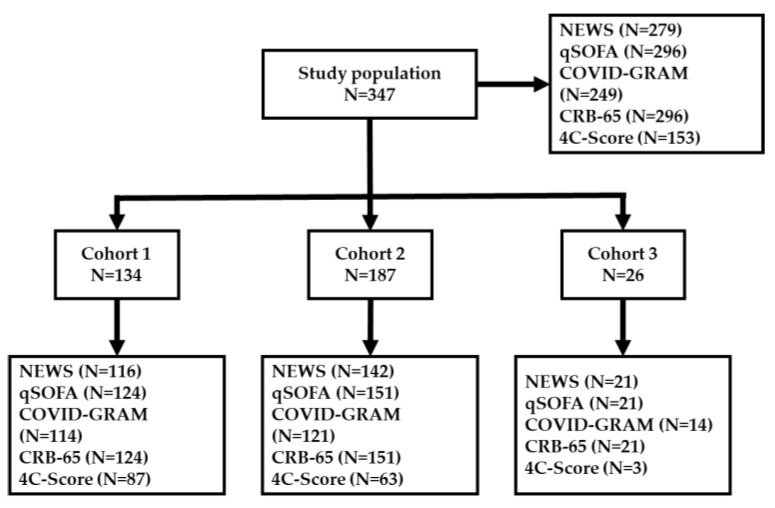
Overview of score calculation. Abbreviations: *NEWS*, National Early Warning Score; *qSOFA*, quick sequential organ failure assessment; *COVID*, coronavirus disease; *GRAM*, Guangzhou Institute of Respiratory Health Calculator at Admission; *CRB*, confusion, respiratory rate, blood pressure.

**Figure 2 jpm-12-01775-f002:**
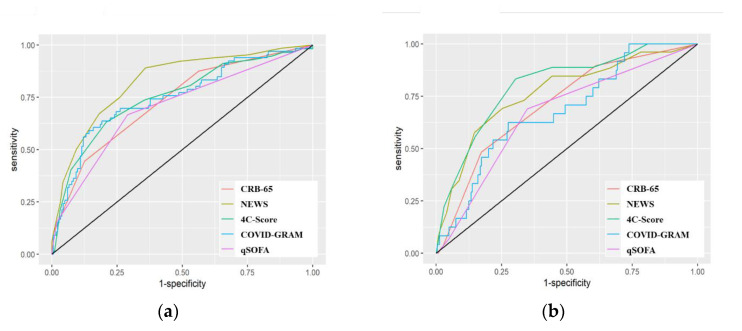
Plots of *area under the receiver operating characteristics* of the scores. (**a**) ICU admission (**b**) and in-hospital mortality. Abbreviations: *CRB*, confusion, respiratory rate, blood pressure; *NEWS*, National Early Warning Score; *COVID*, coronavirus disease; *GRAM*, Guangzhou Institute of Respiratory Health Calculator at Admission; *qSOFA*, quick sequential organ failure assessment.

**Table 1 jpm-12-01775-t001:** Definition of cohorts.

	Cohort 1	Cohort 2	Cohort 3
Admission period	1 March 2020–30 June 2020	1 July 2020–7 March 2021	8 March 2021–30 May 2021
Variants	High prevalence of wild type	High prevalence of B.1.177	High prevalence of alpha variant
Specific therapy	None	Remdesivir, steroids	Remdesivir, steroids
Vaccination	None	Completely vaccinated: 0.9–3.3% * (vaccination available since 26 December 2020)	Completely vaccinated: 6.9–27.4% *

* Vaccination rates according to [16].

**Table 2 jpm-12-01775-t002:** Parameter of risk scores.

qSOFA	CRB-65	NEWS	COVID-GRAM	4C-Score
Respiratory rate Consciousness Blood pressure	Confusion Respiratory rate Blood pressure Age	Respiratory rate Oxygen saturation Suppl. oxygen Temperature Blood pressure Heart rate Consciousness	X-ray abnormality Age Hemoptysis Dyspnea Unconsciousness Number of comorbidities Cancer history Neutrophil/lymphocytes Lactate dehydrogenase Direct bilirubin	Age Sex at birth Number of comorbidities Respiratory rate Oxygen saturation Glasgow Coma Scale Urea CRP level

**Table 3 jpm-12-01775-t003:** Characteristics of study population.

Characteristics	Overall No (%) or Mean (Q1–Q3)	First Cohort No (%) or Mean (Q1–Q3)	Second Cohort No (%) or Mean (Q1–Q3)	Third Cohort No (%) or Mean (Q1–Q3)	*p*-Value *
Study population	347 (100)	134 (38.6)	187 (53.9)	26 (7.5)	-
In-hospital mortality	32 (9.2)	23 (17.2)	9 (4.8)	0 (0)	0.044
ICU admission	82 (23.6)	50 (37.3)	30 (16.0)	2 (7.7)	0.002
Average age	65.4 (57.0–78.0)	67.0 (58.5–80.0)	65.1 (57.0–76.0)	59.6 (51.8–68.3)	1
Women	153 (44.1)	57 (42.5)	84 (44.9)	12 (46.2)	1
COVID-19 vaccination	1 (0.3)	0 (0)	0 (0)	1 (3.8)	-
Respiratory disease	72 (20.7)	22 (16.4)	44 (23.5)	6 (23.1)	1
Cardiovascular disease	246 (70.9)	99 (73.9)	131 (70.1)	16 (61.5)	1
Diabetes	82 (23.6)	32 (23.9)	46 (24.6)	4 (15.3)	1
Hypertension	204 (58.8)	86 (64.2)	106 (56.7)	12 (46.2)	1
Liver disease	27 (7.8)	10 (7.5)	12 (6.4)	5 (19.2)	1
Chronic kidney disease	41 (11.8)	12 (9.7)	25 (13.4)	3 (11.5)	1
HIV or AIDS	0 (0)	0 (0)	0 (0)	0 (0)	-
Organ transplantation	11 (3.2)	4 (3.0)	6 (3.2)	1 (3.8)	1
Malignancy (active)	45 (13.0)	9 (6.7)	34 (18.2)	2 (7.7)	0.236
Malignancy (history)	29 (8.4)	20 (14.9)	7 (3.7)	2 (7.7)	0.070
Neurological conditions	79 (22.8)	29 (21.6)	44 (23.5)	6 (23.1)	1
COVID-19 findings	173 (49.9)	82 (61.2)	79 (42.2)	12 (46.2)	0.272
Oxygen saturation (%)	93.4 (92.0–97.0)	91.9 (90.0–96.0)	94.4 (93.0–97.0)	94.2 (93.0–97.0)	0.034
Heart rate (/min)	86.5 (74.0–96.0)	88.9 (76.0–100.0)	85.1 (72.0–94.0)	83.4 (69.0–91.0)	1
Respiratory rate (/min)	21.1 (16.0–24.0)	22.4 (17.0–26.0)	20.6 (16.3–24.0)	17.8 (15.0–20.0)	1
Syst. Blood pressure (mmHg)	132.0 (118.0–145.0)	129.3 (110.0–140.0)	133.8 (120.0–148.0)	127.1 (116.3–138.3)	1
Temperature (°C)	37.1 (36.4–37.7)	37.3 (36.5–38.1)	36.8 (36.2–37.4)	37.5 (36.8–38.3)	0.036
GCS	14.8 (15.0–15.0)	14.8 (15.0–15.0)	14.9 (15.0–15.0)	14.8 (15.0–15.0)	1
Dyspnea	184 (53.0)	71 (53.0)	103 (55.1)	10 (38.5)	1
Cough	179 (51.6)	84 (62.7)	88 (47.1)	7 (26.9)	0.751
Fever	181 (52.2)	86 (64.2)	80 (42.8)	15 (57.7)	0.026
Leukocyte count (/µL)	6828.2 (4355.0–8355.0)	6586.8 (4385.0–7970.0)	7064.0 (4305.0–8575.0)	6389.1 (4543.0–7665.0)	1
Neutrophil count (10^3^/µL)	5.2 (3.0–6.4)	4.9 (3.1–6.3)	5.4 (2.9–6.8)	5.3 (3.4–6.4)	1
Lymphocyte count (10^3^/µL)	1.1 (0.6–1.2)	1.0 (0.6–1.1)	1.3 (0.6–1.3)	1.0 (1.0–1.2)	1
Urea (mmol/L)	49.5 (27.0–62.0)	47.1 (24.8–62.0)	52.2 (28.0–65.0)	43.5 (29.0–56.8)	1
CRP (mg/dL)	8.0 (1.0–11.9)	8.9 (2.6–13.3)	7.7 (1.7–11.2)	6.3 (1.9–11.5)	1
Bilirubin (mg/dL)	0.7 (0.4–0.8)	0.7 (0.5–0.9)	0.6 (0.4–0.8)	0.6 (0.4–0.8)	0.507
LDH (U/L)	346.6 (229.8–378.5)	328.6 (229.5–389.0)	361.4 (225.0–365.5	344.0 (240.0–411.0)	1
Steroids	146 (42.1)	24 (17.9)	103 (59.9)	19 (73.1)	-
Remdesivir	131 (37.8)	0 (0)	112 (55.1)	19 (73.1)	-

* Bonferroni–Holm adjusted *p*-values for the comparison of the first and second cohorts.

## Data Availability

The data presented in this study are available on request from the corresponding author. Since informed patient consent was waived according to German law, he data are not publicly available.

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
