# Peer review of "External Validation of COVID-19 Risk Scores during Three Waves of Pandemic in a German Cohort—A Retrospective Study"

_jpm, 2022, doi:10.3390/jpm12111775_

Round 1
Reviewer 1 Report
I was very pleased to read the original research manuscript entitled: NEWS in COVID-19 risk scores-external validation of international COVID-19 risk scores during three waves of pandemic in a German cohort-a retrospective study. The topic is interesting as it assess the performance of several preexisting risk-scores such as NEWS, CRB-65, qSOFA and COVID-19 specific scores (COVID-GRAM, 4C-Mortality score) for the prediction of in-hospital mortality and ICU-admission in the same cohort of German patients.
Minor points:
1. The title is a bit long and difficult to follow and I suggest you to revise it.
2. Regarding vaccinated patients: is there any data regarding the type of vaccine administrated? If information is available please include in text.
3. Please mention/discuss the overall national in-hospital mortality and ICU-admission rate in the studied COVID-19 waves.
4. Please revise the patents section. If nor necessary please delete.
Author Response
- The title is a bit long and difficult to follow and I suggest you to revise it.
Thank you very much for this advice – we followed your suggestion:
External Validation of COVID-19 risk scores during three waves of pandemic in a German cohort - a retrospective study
- Regarding vaccinated patients: is there any data regarding the type of vaccine administrated? If information is available please include in text.
In our cohort we had only one vaccinated patient as described in table 3 and in results 3.1. Unfortunately the type of vaccine is unknown. Since it was only one patient we thought the statistical significance of the details is minor. But we included the information in the results part now. The numbers shown in Table 1 are referring to overall national vaccination rates as mentioned in (16). We mentioned these to show the possible indirect impact on the selection of patients requiring hospitalisation.
- Please mention/discuss the overall national in-hospital mortality and ICU-admission rate in the studied COVID-19 waves.
We thank you for this suggestion which improves the discussion. Unfortunately central national data that differentiate the waves are not systematically available, but two studies on health insurance data from Germany provide the information. We added the information in the discussion, please see line 252-267 and reference 26. and 27 - please see the attachment in track change mode.
- Please revise the patents section. If nor necessary please delete.
Thank you for your mindfulness – we missed that point. We deleted this section

Reviewer 2 Report
1. By comparing the five risk scoring methods, the following conclusions were drawn: NEWS and 4C-Score performed best for the outcomes ICU admission and in-hospital mortality, the easy to use bedside score NEWS has proven to identify patients at risk for critical illness, this is of great significance for the precise treatment of severe COVID-19 patients and the reduction of mortality. However, patients infected with low mortality virus strains such as Omicron affect the discriminative ability of these scoring methods, and new risk scoring methods need to be developed.
2. A period is missing at the end of line 298, 307, 308, 316 and 323.
Author Response
Dear Reviewer, thank you for your kind review.
We filled in the missing periods and thank you for your mindfullness.